# Long-Term Hydroxychloroquine Therapy and Risk of Coronary Artery Disease in Patients with Systemic Lupus Erythematosus

**DOI:** 10.3390/jcm8060796

**Published:** 2019-06-05

**Authors:** Deng-Ho Yang, Pui-Ying Leong, Sung-Kien Sia, Yu-Hsun Wang, James Cheng-Chung Wei

**Affiliations:** 1Division of Rheumatology/Immunology/Allergy, Department of Internal Medicine, Taichung Armed-Forces General Hospital, Taichung 411, Taiwan; deng6263@ms71.hinet.net; 2Department of Medical Laboratory Science and Biotechnology, Central Taiwan University of Science and Technology, Taichung 406, Taiwan; 3Division of Rheumatology/Immunology/Allergy, Department of Internal Medicine, Tri-Service General Hospital, National Defense Medical Center, Taipei 114, Taiwan; 4Institute of Medicine, Chung Shan Medical University, Taichung 402, Taiwan; fiona.leong@gmail.com (P.-Y.L.); peterssk1@gmail.com (S.-K.S.); 5Department of Medicine, Chung Shan Medical University Hospital, Taichung 402, Taiwan; 6Department of Medical Research, Chung Shan Medical University Hospital, Taichung 402, Taiwan; cshe731@csh.org.tw; 7Graduate Institute of Integrated Medicine, China Medical University, Taichung 402, Taiwan

**Keywords:** Hydroxychloroquine, cardiovascular disease, atherosclerosis, inflammation, stroke, lupus

## Abstract

Systemic lupus erythematosus (SLE) is a chronic systemic inflammatory disease associated with a high prevalence of cardiovascular disease (CVD). Hydroxychloroquine (HCQ) is commonly used to control disease activity in patients with SLE. We evaluated its potential additional therapeutic effect for reducing CVD in SLE patients. We conducted a retrospective cohort study, in which one million participants were sampled from 23 million beneficiaries and data were collected from 2000 to 2013. In total, 826 SLE patients receiving HCQ medication were included after exclusion for previous CVD. The total number of SLE patients was 795 after follow-up for more than one year. After adjusting for chronic comorbidity, a significantly decreased hazard ratio (HR) for coronary artery disease (CAD) was found among SLE patients with a high usage of HCQ for at least 318 days (HR = 0.31, 95% confidence interval, CI: 0.12–0.76). A low HR for CAD was observed in SLE patients with a high cumulative dose of at least 100,267 mg HCQ (HR = 0.25, 95% CI: 0.09–0.66). However, there was no significant lowering of the HR for stroke. Long-term HCQ therapy decreases the HR of CVD in SLE patients. The cardiovascular protective effect of HCQ therapy was associated with decrease in CAD, but not stroke.

## 1. Introduction

Systemic lupus erythematosus (SLE) is a systemic inflammatory disease with multiple organ involvement including skin, bone, heart, brain, kidney, and liver [1]. Cardiovascular events are one of the major causes of mortality or morbidity in patients with SLE [2]. Further, these patients have a higher prevalence of cardiovascular events compared with the general population [3,4]. Cardiac dysfunction in SLE may be symptomatic or subclinical. Subclinical cardiac presentations include mitral regurgitation, pericardial effusion, aortic regurgitation, tricuspid regurgitation, and ventricular hypertrophy [5]. Cardiac valvular abnormalities may be found in SLE patients [6]. Elevation of the level of circulating anti-phospholipid antibodies including lupus anticoagulant, anticardiolipin antibodies, and anti-β 2 glycoprotein antibodies is usually observed during the disease course of SLE [7]. Thrombosis-related chronic morbidity may progress in SLE patients. Symptomatic presentations of ischemic heart disease are higher in patients with SLE than in the general population, and SLE is associated with the progression of ischemic heart disease [8]. A meta-analysis revealed a higher prevalence of increased carotid intima and media thickness and carotid plaques in SLE patients [9]. Other thrombosis-related cardiovascular diseases (CVD), such as stroke, also occur more frequently in SLE patients [10]. The traditional risk factors for CVD include diabetes, hyperlipidemia, hypertension, obesity, family history of coronary heart disease, and cigarette smoking. These are also the major risk factors for CVD development in SLE [11]. Systemic inflammation, presentation of autoantibody-related immune responses, and gene variant-related immune dysregulation may be involved in CVD pathogenesis in SLE patients [12,13,14]. The increased risk of CVD in SLE patients may be the result of the interaction between traditional CVD risk factors and SLE related risk factors (including systemic inflammation, production of autoantibodies and medical treatment). In patients with SLE, steroid administration is performed to control disease activity and may be associated with CVD risk. However, the role of immunomodulation agents in the pathogenesis of thrombosis related disease is still unclear. The usual immunomodulation agents in SLE include hydroxychloroquine (HCQ), azathioprine, methotrexate, and mycophenolate. Adequate immune suppression or regulation of SLE to control disease activity can be an important issue for the progression of CVD. The use of therapeutic corticosteroids is one risk factor for CVD in rheumatoid arthritis patients [15]. One previous hospital study suggested that prolonged medication with HCQ plus low dose aspirin has a thromboprotective effect in SLE patients [16]. Meta-analysis data suggest a possible clinical effect of HCQ in decreasing insulin resistance and the incidence of CVD [17]. Medication with chloroquine or HCQ is associated with decreasing risk of CVD in patients with rheumatic diseases in most observational studies [18]. The pharmacologic effect of HCQ in the prevention of CVD seems to involve not only an anti-inflammatory response but other responses as well. Therefore, we aimed to evaluate the effect of long-term HCQ therapy in the development of CVD, including coronary artery disease (CAD) and stroke, in SLE patients.

## 2. Experimental Section

### 2.1. Methodology

We conducted a retrospective cohort study by using the National Health Insurance Research Database (NHIRD), which enrolls almost 99% of the population in Taiwan. The dataset consists of all claims in health care, including medical visits, emergency care, and hospitalization. International Classification of Diseases-9-Clinical Modification (ICD-9-CM) system was used to code the diseases. One million participants were sampled from the 23 million beneficiaries, and data were collected from 2000 to 2013. The data in the NHIRD were encrypted, and this study was approved by the Institutional Review Board of Chung Shan Medical University Hospital. The study population consisted of patients newly diagnosed with SLE (ICD-9-CM = 710.0), aged 20 years or more, from 2000 to 2012. Administration of HCQ within 6 months of diagnosis was necessary for inclusion of these SLE patients. A total of 1072 SLE patients treated with HCQ were enrolled. To confirm new-onset disease, we excluded those with CVD diagnosed before the index date. Therefore, 826 SLE patients were included after exclusion for previous CVD. The index date was the first date of usage of HCQ. The outcome variable was defined as a diagnosis of CVD including CAD (ICD-9-CM = 410–414) and stroke (ICD-9-CM = 430–437). Additionally, more than three outpatient visits or a hospitalization due to CVD were required for inclusion. Patients were followed up with until the occurrence of CVD, the date of 31 December 2013 was reached, or withdrawal from the national insurance system, whichever occurred first. The total number of SLE patients was 795 after follow-up for more than one year. The study flow chart to identify these SLE patients is shown in Figure 1.

The cumulative doses of HCQ were calculated within one year from index date. The medication possession ratio (MPR), which was defined as the cumulative days of HCQ in one year, was also used to estimate the adherence to medication [19]. We identified baseline characteristics including age, sex, hypertension (ICD-9-CM = 401–405), hyperlipidemia (ICD-9-CM = 272.0–272.4), chronic liver disease (ICD-9-CM = 571), chronic kidney disease (ICD-9-CM = 585), chronic obstructive pulmonary disease (COPD, ICD-9-CM = 490–492, 494, 496), and diabetes (ICD-9-CM = 250). These comorbidities were defined as occurring before or within one year of the index date and resulting in more than three outpatient visits or at least one hospitalization.

### 2.2. Statistical Analysis

The Kaplan-Meier analysis was used to estimate the cumulative incidence of CVD across the HCQ groups, and the log-rank test was used to evaluate the significance. The Cox proportional hazard model was used to estimate the hazard ratio of cardiovascular disease in relation to HCQ medication and was adjusted for potential confounding variables. We used statistical software SPSS, version 18.0 (SPSS Inc., Chicago, IL, USA). A *p* value less than 0.05 was considered to indicate significance.

## 3. Results

### 3.1. SLE Patient Characteristics

A total of 826 patients with SLE were included. Female sex predominated with a ratio of 9 to 1 (female to male). The average age was 40 years. Different chronic diseases were found in these patients. The most common comorbid diseases were hypertension (6.8%) and chronic liver disease (5.3%). Duration of HCQ administration from the index date was differentiated for three groups: low (<105 days), medium (105–318 days), and high (≥318 days). Three groups including low (<30,800 mg), medium (30,800–100,267 mg) and high dosage (≥100,267) of HCQ were evaluated by the cumulative dose. MPR of HCQ from index date was separated for three groups: low (<0.29), medium (0.29–0.87) and high (≥0.87). The demographic characteristics of SLE patients are shown in Table 1.

### 3.2. A Decreased Hazard Ratio (HR) for CVD in SLE Patients with High Usage, High Cumulative Dose of HCQ or High MPR of HCQ

A higher HR (6.29, 95% CI: 2.83–14.02) for CVD was found in the old age group (≥45 years old) when compared with the group of age <30 years. Hypertension was also a risk for progression of CVD (HR 3.08, 95% CI: 1.65–5.74). There was no significant increase in HR in other chronic diseases including hyperlipidemia, chronic liver disease, COPD and diabetes (See Table 2). SLE patients with high usage of HCQ (≥318 days) had significant reduction of HR when compared with low usage (<105 days). The HR was 0.38 (95% CI: 0.21–0.70) after adjusting for age, sex, hypertension, hyperlipidemia, chronic liver disease, COPD, and diabetes. A high cumulative HCQ dose of more than 100,267 mg had a lower HR when compared with low dose (HR = 0.42, 95% CI: 0.23–0.77). The hazard ratios of developing CVD in different usages and cumulative doses of HCQ is shown in Table 2. The cumulative probability of CVD and CAD had significantly decreased with higher usage of HCQ for more than 318 days (Figure 2). A high MPR of HCQ (≥0.87) was associated with decreasing HR for CVD compared with low MPR (HR = 0.38, 95% CI: 0.21–0.70), as shown in Table 3.

### 3.3. Decreased HR for CAD in SLE Patients with High Usage of HCQ, High Cumulative Dose of HCQ, or High MPR of HCQ

After adjusting for chronic comorbidity, a significant decrease in HR for CAD was found in SLE patients with a high compared with low usage of HCQ (HR = 0.31, 95% CI: 0.12–0.76). A low HR for CAD was observed with a high cumulative dose of HCQ when compared with low (HR = 0.25, 95% CI: 0.09–0.66). A low HR for CAD was also observed with a high MPR of HCQ when compared with low (HR = 0.31, 95% CI: 0.12–0.76). However, there was no significant lowering of HR for stroke in SLE patients with high usage, high cumulative dose, or high MPR of HCQ. The different HRs for CAD and stroke in SLE patients treated with HCQ is shown in Table 4.

## 4. Discussion

SLE is a systemic inflammatory disease involving different circulating autoantibodies. Some, such as anti-cardiolipin antibodies, may be associated with risk for progression to thrombosis-related disease. During the disease course of SLE, persistent systemic inflammation may also be associated with development of CAD. Additionally, administration of steroid may increase risk of CVD. Therefore, a higher incidence of CVD is observed in SLE patients when compared with the general population. In our study, a longer duration of HCQ medication decreased the HR for CVD (Figure 2). HCQ is one of the common immunosuppressive agents used for treatment of SLE. The effects of HCQ in SLE include reduction of disease flares, steroid dosage, organ damage, and anti-phospholipid antibodies [20]. The decrease in circulating anti-phospholipid antibodies is associated with reduction of thrombotic events.

A significant reduction of HR for CVD was observed in SLE patients with high dosage HCQ for more than 318 days (HR = 0.38, 95% CI: 0.21–0.70). A single hospital study showed long-term use of HCQ with low-dose aspirin protecting against thrombosis in patients with SLE [16].

In the pathogenic inflammation of SLE, production of different circulating autoantibodies may induce immune complex deposition in different tissues and organs, leading to tissue destruction [21]. In SLE, thrombosis-related antiphospholipid autoantibodies include anti-cardiolipin antibodies and anti-beta-2-GPI antibodies. In the mouse model of antiphospholipid syndrome, HCQ reverses the prothrombotic state by reduction of inflammation and prevention of endothelial dysfunction [22,23]. In vivo, HCQ can protect from inflammation and prothrombotic signaling pathways by inhibiting endosomal NADPH oxidase [24]. A significant reduction of circulating tissue factor levels is observed in patients with antiphospholipid antibodies, after three months of HCQ therapy [25]. The use of HCQ can inhibit production of various antiphospholipid antibodies and prevent dysfunction of endothelial cells. Modulation and regulation of SLE related systemic inflammation is another effect of HCQ.

In our study, a significant reduction of HR for CVD was observed in patients with a high cumulative dose of HCQ (≥100,267 mg) or a high MPR of HCQ (Table 1 and Table 2). However, the thrombotic protection of HCQ was not always observed. Sufficient long-term medication (≥318 days) or an adequate cumulative dose (≥100,267 mg) of HCQ should be achieved for its preventive effect against CVD to develop.

A higher prevalence of accelerated and early coronary artery arteriosclerosis is observed in SLE patients and SLE is an independent risk factor for CVD [11,26]. The protection of HCQ in the early atherosclerosis of SLE may be associated with inhibition of toll-like receptor signaling, proinflammatory cytokine production, T-cell and monocyte activation, oxidative stress, and endothelial dysfunction [27]. Therefore, HCQ has a clear effect against accelerated atherosclerosis of the coronary arteries.

We found that high usage or cumulative dose or MPR of HCQ caused a significant reduction of HR for CAD when compared with low dose (Table 4). Hyperlipidemia is a risk factor for CVD. Lower low-density lipoprotein and total cholesterol levels are associated with HCQ treatment in RA [17,28]. Significantly lower low-density lipoprotein is observed in patients treated with HCQ medication for at least three months [29]. A long duration of HCQ therapy is important for protection against CVD by improving lipid profiles. We showed that long-term medication of at least 318 days is associated with a decreasing HR for CVD (Figure 2). Lower common carotid artery intima media thickness is found after HCQ therapy in patients with SLE [30]. HCQ use is associated with lower low-density lipoprotein levels but not carotid artery intima media thickness in lupus nephritis patients [31]. Hydroxychloroquine has no significant protective effect on ischemic stroke in rheumatoid arthritis patients [32]. Whether HCQ therapy in SLE decreases risk of stroke is still controversial. In our study, a protective effect of HCQ therapy in the development of stroke was not observed.

Higher doses of prednisolone (>7.5 mg/day) are associated with CVD in SLE patients [33]. A significant reduction of prednisolone dose is associated with HCQ therapy in Japanese patients with SLE [34]. However, medication of steroid, other immunomodulation agents except HCQ, antihypertensive drugs, anticoagulant and biologics were not included to evaluate in this study. These treatments may influence the risk of CVD in SLE patients, and was a limitation in our study. When SLE patients receive adequate HCQ therapy, decreasing the dose of steroid may decrease the risk of CVD. The adverse events of HCQ therapy include retinopathy, cutaneous hyperpigmentation, myopathy, neurocardiomyopathy, acute generalized exanthematous pustulosis, agranulocytosis, anemia, aplastic anemia, leukopenia, and thrombocytopenia [35]. Duration of HCQ use, daily HCQ dose, and chronic kidney disease are associated with the incidence of HCQ related retinal toxicity [36]. High usage and long duration of HCQ therapies could have several and irreversible side effects that are frequently found in SLE patients and may be a limitation or possible handicap of long-term usage of HCQ for preventing CVD. Thus, further studies (interventional) could be interesting in order to clarify if the risks of high usage and dosage of HCQ for preventing CVD are sustainable and do not suppose a major risk for other health conditions. In the general population, a multicenter clinical trial is being conducted to evaluate whether treatment with HCQ reduces risk of recurrent CVD in patients with CAD [37].

## 5. Conclusions

In conclusion, a long duration of HCQ therapy, with an adequate dose, decreases the HR for CVD in SLE patients. The cardiovascular protective effect of HCQ therapy was associated with decrease in CAD, but not stroke, in our study. The possible mechanisms include inhibition of systemic inflammation, regulation of antiphospholipid antibodies, protection from endothelial dysfunction, reduction of steroid dose, and improved lipid profiles. This study demonstrated a significant, dose-related association between HCQ therapy and reduced risk of CAD in patients with SLE. HCQ may have a role in the treatment of SLE not only to control disease activity but also to reduce the risk of developing CAD. Long term and high/adequate doses of treatment with HCQ may have protective effects against CAD in SLE patients. Therefore, using longer duration of HCQ therapy for more than one year may be suggested among the SLE patients with high risk of progressive CAD due to the cardiovascular protective effect of HCQ. Adequate accumulative dose of HCQ for more than 100,267 mg had significant decreasing HR of CAD. At the same time, monitor and follow up the toxicity of HCQ should be always keep in mind to prevent progressive adverse events. Further studies are needed to evaluate the real role of HCQ as a preventive agent of CAD in the future.

## Figures and Tables

**Figure 1 jcm-08-00796-f001:**
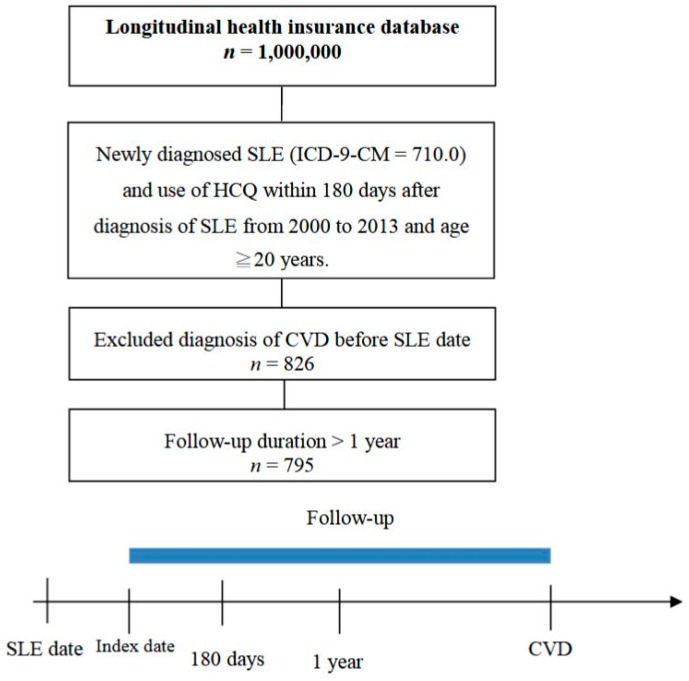
Study flow chart of the SLE patients with progression of CVD. SLE: Systemic lupus erythematosus; CVD: cardiovascular disease; HCQ: Hydroxychloroquine; ICD-9-CM: International Classification of Diseases-9-Clinical Modification.

**Figure 2 jcm-08-00796-f002:**
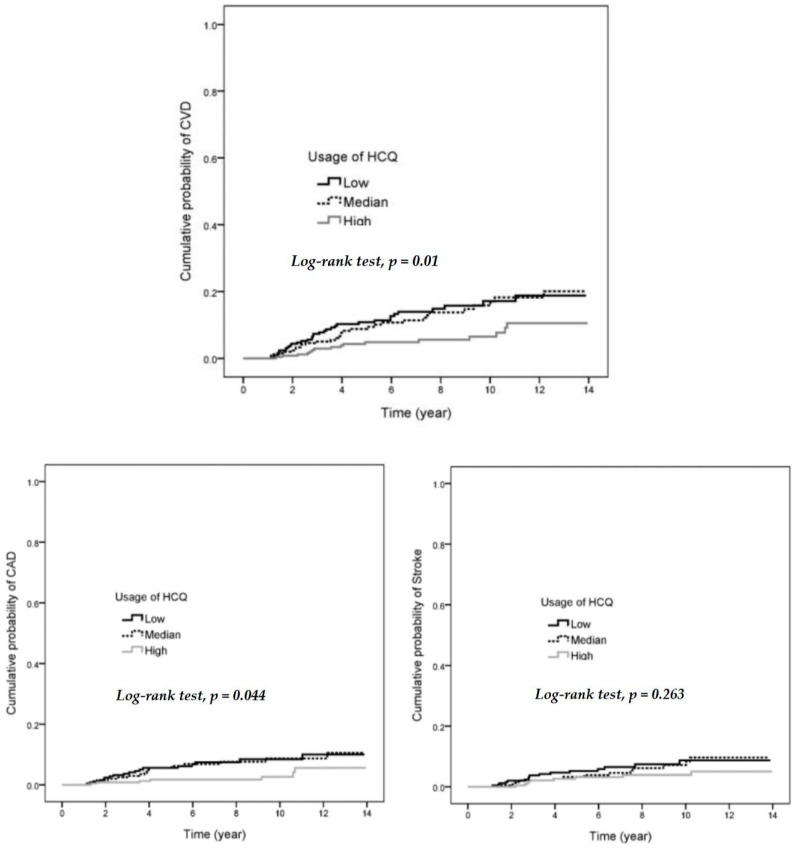
Cumulative probability of CVD, CAD, and stroke in low, median, and high usage of HCQ. CAD: coronary artery disease.

**Table 1 jcm-08-00796-t001:** Demographic characteristics of SLE patients.

	*n*	%
Age		
<30	208	26.2
30–45	318	40.0
≥45	269	33.8
Mean ± SD	40 ± 13.3	
Sex		
Female	715	89.9
Male	80	10.1
Hypertension	54	6.8
Hyperlipidemia	22	2.8
Chronic liver disease	42	5.3
Chronic kidney disease	4	0.5
COPD	26	3.3
Diabetes	20	2.5
Usage of HCQ (days)		
Low (<105)	264	33.2
Median (105–318)	265	33.3
High (≥318)	266	33.5
Cumulative HCQ dose (mg)		
Low (<30,800)	263	33.1
Median (30,800–100,267)	267	33.6
High (≥100,267)	265	33.3
MPR of HCQ		
Low (<0.29)	264	33.2
Median (0.29–0.87)	265	33.3
High (≥0.87)	266	33.5

COPD: chronic obstructive pulmonary disease; HCQ: hydroxychloroquine; MPR: Medication possession ratio; SLE: Systemic lupus erythematosus; SD: Standard deviation.

**Table 2 jcm-08-00796-t002:** HRs of developing CVD according to age, sex, comorbidity, usage, and cumulative dose of HCQ in the patients with SLE.

	CVD Event	Observed Person–Years	Incidence (/1000 Person–Years)	HR	95% CI	Adjusted HR ^†^	95% CI
Age (years)							
<30	7	1623	4.3	1		1	
30–45	18	2146	8.4	1.95	0.81–4.66	1.85	0.77–4.44
≥45	56	1726	32.4	7.53	3.43–16.53	6.29	2.83–14.02
Sex							
Female	70	5023	13.9	1		1	
Male	11	472	23.3	1.68	0.89–3.18	1.24	0.65–2.38
Hypertension	16	279	57.3	4.55	2.62–7.88	3.08	1.65–5.74
Hyperlipidemia	4	132	30.4	2.11	0.77–5.75	0.68	0.23–2.01
Chronic liver disease	8	280	28.6	2.03	0.98–4.21	1.48	0.68–3.19
COPD	4	163	24.5	1.72	0.63–4.69	0.68	0.23–2.02
Diabetes	3	125	24.1	1.65	0.52–5.22	0.67	0.20–2.26
Usage of HCQ (days)							
Low (<105)	34	1745	19.5	1		1	
Median (105–318)	31	1787	17.3	0.90	0.55–1.47	0.76	0.46–1.25
High (≥318)	16	1962	8.2	0.42	0.23–0.76	0.38	0.21–0.70
Cumulative HCQ dose (mg)							
Low (<308,00)	33	1750	18.9	1		1	
Median (308,00–100,267)	30	1754	17.1	0.92	0.56–1.51	0.92	0.56–1.52
High (≥100,267)	18	1991	9.0	0.48	0.27–0.86	0.42	0.23–0.77

HR: hazard ratio; CI: confidence interval; ^†^ Adjusted for age, sex, hypertension, hyperlipidemia, chronic liver disease, COPD, and diabetes.

**Table 3 jcm-08-00796-t003:** High MPR of HCQ was associated with decreasing HR of CVD in SLE patients.

	N	CVD Event	HR	95% CI	Adjusted HR ^†^	95% CI
MPR of HCQ						
Low (<0.29)	264	34	1		1	
Median (0.29–0.87)	265	31	0.90	0.55–1.47	0.77	0.47–1.27
High (≥0.87)	266	16	0.42	0.23–0.76	0.38	0.21–0.70

^†^ Adjusted for age, sex, hypertension, hyperlipidemia, chronic liver disease, COPD and diabetes. MPR: Medication possession ratio.

**Table 4 jcm-08-00796-t004:** HRs of CAD and stroke in SLE patients treated with HCQ.

	*n*	Event	HR	95% CI	Adjusted HR ^†^	95% CI
**CAD**						
Usage of HCQ (days)						
Low (<105)	264	18	1		1	
Median (105–318)	265	17	0.95	0.49–1.84	0.71	0.36–1.40
High (≥318)	266	7	**0.36**	**0.15–0.87**	**0.31**	**0.12–0.76**
Cumulative HCQ dose (mg)						
Low (<30,800)	263	17	1		1	
Median (30,800–100,267)	267	19	1.16	0.60–2.23	1.14	0.58–2.24
High (≥100,267)	265	6	**0.32**	**0.13–0.81**	**0.25**	**0.09–0.66**
MPR of HCQ						
Low (<0.29)	264	18	1		1	
Median (0.29–0.87)	265	17	0.95	0.49–1.84	0.71	0.36–1.40
High (≥0.87)	266	7	**0.36**	**0.15–0.87**	**0.31**	**0.12–0.76**
Stroke						
Usage of HCQ (days)						
Low (<105)	264	16	1		1	
Median (105–318)	265	14	0.88	0.43–1.80	0.82	0.39–1.70
High ≥318)	266	9	0.52	0.23–1.17	0.51	0.22–1.16
Cumulative HCQ dose (mg)						
Low (<30,800)	263	16	1		1	
Median (30,800–100,267)	267	11	0.68	0.32–1.47	0.68	0.31–1.46
High (≥100,267)	265	12	0.69	0.33–1.46	0.67	0.31–1.42
MPR of HCQ						
Low (<0.29)	264	16	1		1	
Median (0.29–0.87)	265	14	0.88	0.43–1.80	0.82	0.39–1.70
High (≥0.87)	266	9	0.52	0.23–1.17	0.51	0.22–1.16

Bold font represents statistical significance (*p* < 0.05). ^†^ Adjusted for age, sex, hypertension, hyperlipidemia, chronic liver disease, COPD and diabetes. MPR: Medication possession ratio.

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
