# Peer review of "Long-Term Hydroxychloroquine Therapy and Risk of Coronary Artery Disease in Patients with Systemic Lupus Erythematosus"

_jcm, 2019, doi:10.3390/jcm8060796_

Reviewer 1 Report

This is a very interesting and well written study about the long term effects of HCQ in vascular disease in SLE patients. However, it is not a novel one. It would be interesting if this article would be supported with other information like HCQ safety in the long term/high dose treatment in SLE patients.

Lines 43 – 44 and lines 45. Please add a reference for each statement.

Lines 47 it does not seems to be clear the difference between symptomatic or asymptomatic CVD in SLE...I suggest to change the word “subclinical” for the word “asymptomatic” to clearly state which conditions are the asymptomatic or subclinical ones

Lines 50-52…Please add a reference for this statement.

Line 59. You are stating here that traditional CVD risk factors (Framingham) are more important that other factors (inflammation, autoantibodies, medications) in the development/progression of CVD in lupus patients. It could be possible but I suggest that you add a reference for justifying this. If you find it it would be OK. But, if this is not clear, it would be better to reshape the statement saying that the increased risk of CVD in SLE patients may be the result of the interaction between traditional or classical CVD risk factors and SLE related factors (inflammation, autoantibodies, medical treatment...etc)

Line 84 ICD-9-CM Is it an acronym? I have not seen it before in the text. Please define for a better interpretation.

Line 129. Did you assess information regarding other medications used in these patients ? (esteroids, immunossupressors, antiagregants, antihypertensive, statins)...this should be stated and deeply explained. This may affect results, and also they should be considered as confounding factors.

Line 270. High usage and long duration of HCQ therapies could have several and irreversible side effects that are frequently found in SLE patients. It would be of interest if you address this fact as limitations or a possible handicap of the high use of HCQ for preventing CVD. Thus, further studies (interventional) could be interesting in order to clarify if the risks of high usage and dosage of HCQ for preventing CVD is sustainable and do not suppose a major risk for other health conditions

Other possibility that would enhance your manuscripto could be showing, specially in the long term treatment group,  if there were reported other medical complications (visual/hearing problems, hair loss) that could be consequence of the treatment with HCQ.  This also would enhance your paper making it different to previous articles.

Line 281. “aggressive treatment”…this sounds like  a very hard adjective...i suggest also to specify what you are meaning to say....dont you think it would be better to stay "long term and high/adequate doses..."

Author Response

This is a very interesting and well written study about the long term effects of HCQ in vascular disease in SLE patients. However, it is not a novel one. It would be interesting if this article would be supported with other information like HCQ safety in the long term/high dose treatment in SLE patients.

Lines 43 – 44 and lines 45. Please add a reference for each statement.

Response:

We have added reference as below: Systemic lupus erythematosus (SLE) is a systemic inflammatory disease with multiple organ involvement including skin, bone, heart, brain, kidney, and liver [1]. Cardiovascular events are one of the major causes of mortality or morbidity in patients with SLE [2]. See line 43-45.

Lines 47 it does not seems to be clear the difference between symptomatic or asymptomatic CVD in SLE...I suggest to change the word “subclinical” for the word “asymptomatic” to clearly state which conditions are the asymptomatic or subclinical ones.

Response:

We have corrected as below: Cardiac dysfunction in SLE may be symptomatic or subclinical. See line 47.

Lines 50-52…Please add a reference for this statement.

Response:

We have added a reference as below: Elevation of the level of circulating anti-phospholipid antibodies including lupus anticoagulant, anticardiolipin antibodies, and anti-β 2 glycoprotein antibodies is usually observed during the disease course of SLE [7]. See line 50-52.

Line 59. You are stating here that traditional CVD risk factors (Framingham) are more important that other factors (inflammation, autoantibodies, medications) in the development/progression of CVD in lupus patients. It could be possible but I suggest that you add a reference for justifying this. If you find it it would be OK. But, if this is not clear, it would be better to reshape the statement saying that the increased risk of CVD in SLE patients may be the result of the interaction between traditional or classical CVD risk factors and SLE related factors (inflammation, autoantibodies, medical treatment...etc)

Response:

We have added a reference as below: These are also the major risk factors for CVD development in SLE [11]. See line 59.

The increased risk of CVD in SLE patients may be the result of the interaction between traditional CVD risk factors and SLE related risk factors (including systemic inflammation, production of autoantibodies and medical treatment). See line 62-64.

Line 84 ICD-9-CM Is it an acronym? I have not seen it before in the text. Please define for a better interpretation.

Response:

International Classification of Diseases (ICD). Clinical Modification(CM).

We have added as below: International Classification of Diseases-9-Clinical Modification (ICD-9-CM) system was used to code the diseases. See line 83-84.

Line 129. Did you assess information regarding other medications used in these patients ? (esteroids, immunossupressors, antiagregants, antihypertensive, statins)...this should be stated and deeply explained. This may affect results, and also they should be considered as confounding factors.

Response:

We identified baseline characteristics including age, sex, hypertension, hyperlipidemia, chronic liver disease, chronic kidney disease, chronic obstructive pulmonary disease, and diabetes. These comorbidities related CVD were considered in our study. Higher doses of prednisolone (>7.5 mg/day) are associated with CVD in SLE patients [33]. A significant reduction of prednisolone dose is associated with HCQ therapy in Japanese patients with SLE [34]. However, steroid and immunomodulation agents except HCQ were not included to evaluate in our study. Medication of steroid may influence the risk of CVD, and was a limitation in our study. See line 271-275.

Line 270. High usage and long duration of HCQ therapies could have several and irreversible side effects that are frequently found in SLE patients. It would be of interest if you address this fact as limitations or a possible handicap of the high use of HCQ for preventing CVD. Thus, further studies (interventional) could be interesting in order to clarify if the risks of high usage and dosage of HCQ for preventing CVD is sustainable and do not suppose a major risk for other health conditions.

Response:

The adverse events of HCQ therapy including retinopathy, cutaneous hyperpigmentation, myopathy, neurocardiomyopathy, acute generalized exanthematous pustulosis, agranulocytosis, anemia, aplastic anemia, leukopenia, and thrombocytopenia [35]. Duration of HCQ use, daily HCQ dose, and chronic kidney disease are associated with the incidence of HCQ related retinal toxicity [36]. Therefore, HCQ therapy related adverse events may be a possible handicap of long term usage of HCQ for preventing CVD. Thus, further studies could be interesting in order to clarify if the risks of high usage and dosage of HCQ for preventing CVD is sustainable. In the general population, a multicenter clinical trial is being conducted to evaluate whether treatment with HCQ reduces risk of recurrent CVD in patients with CAD [35]. See line 276-284.

Other possibility that would enhance your manuscripto could be showing, specially in the long term treatment group,  if there were reported other medical complications (visual/hearing problems, hair loss) that could be consequence of the treatment with HCQ.  This also would enhance your paper making it different to previous articles.

Response:

Our study used the National Health Insurance Research Database and the long time follow up the adverse events of HCQ was not evaluated in our study.

Line 281. “aggressive treatment”…this sounds like  a very hard adjective...i suggest also to specify what you are meaning to say....dont you think it would be better to stay "long term and high/adequate doses..."

Response:

We have corrected as below: Long term and high/adequate doses treatment with HCQ may have protective effects against CAD in SLE patients. See line 293-294.

Reviewer 2 Report

In this report the authors highlight the value of HCQ on cardiovascular outcomes in SLE population. The authors presented large number of sample size with well balanced events. With paper therapy, it seem that HCQ provided a protective role from cardiac stand point but not cerebrovascular events namely stoke. The analysis has been appropriate. 

I will recommend the discussion to be refined , with first paragraph to focus on the relevant findings here rather that reviewing the literature

Author Response

In this report the authors highlight the value of HCQ on cardiovascular outcomes in SLE population. The authors presented large number of sample size with well balanced events. With paper therapy, it seem that HCQ provided a protective role from cardiac stand point but not cerebrovascular events namely stoke. The analysis has been appropriate.

I will recommend the discussion to be refined , with first paragraph to focus on the relevant findings here rather that reviewing the literature

Response:

We have added a paragraph as below: This study demonstrated a significant, dose-related associated between HCQ therapy and reduced risk of CAD in patients with SLE. HCQ may have a role in the treatment of SLE not only to control disease activity but also to reduce the risk of developing CAD. Long term and high/adequate doses treatment with HCQ may have protective effects against CAD in SLE patients. Therefore, using longer duration of HCQ therapy for more than one year may be suggested among the SLE patients with high risk of progressive CAD due to the cardiovascular protective effect of HCQ. Adequate accumulative dose of HCQ for more than 100,267 mg had significant decreasing HR of CAD. At the same time, monitor and follow up the toxicity of HCQ should be always keep in mind to prevent progressive adverse events. Further studies are needed to evaluate the real role of HCQ as a preventive agent of CAD in the future. See line 290-300.

Round  2

Reviewer 1 Report

Line 324. Please include as limitations also that you did not address information about the use of antihypertensive, antiagregants, statins or biological treatments. This may influence the development of CVD in SLE patients. 

Line 270. High usage and long duration of HCQ therapies could have several and irreversible side effects that are frequently found in SLE patients. It would be of interest if you address this fact as limitations or a possible handicap of the high use of HCQ for preventing CVD. Thus, further studies (interventional) could be interesting in order to clarify if the risks of high usage and dosage of HCQ for preventing CVD is sustainable and do not suppose a major risk for other health conditions.

Author Response

Line 324. Please include as limitations also that you did not address information about the use of antihypertensive, antiagregants, statins or biological treatments. This may influence the development of CVD in SLE patients.

Response:

We have corrected as below: However, medication of steroid, other immunomodulation agents except HCQ, antihypertensive drugs, anticoagulant and biologics were not included to evaluate in this study. These treatments may influence the risk of CVD in SLE patients, and was a limitation in our study. See line 273-275.

Line 270. High usage and long duration of HCQ therapies could have several and irreversible side effects that are frequently found in SLE patients. It would be of interest if you address this fact as limitations or a possible handicap of the high use of HCQ for preventing CVD. Thus, further studies (interventional) could be interesting in order to clarify if the risks of high usage and dosage of HCQ for preventing CVD is sustainable and do not suppose a major risk for other health conditions.

Response:

We have corrected as below: High usage and long duration of HCQ therapies could have several and irreversible side effects that are frequently found in SLE patients and may be a limitation or possible handicap of long term usage of HCQ for preventing CVD. Thus, further studies (interventional) could be interesting in order to clarify if the risks of high usage and dosage of HCQ for preventing CVD is sustainable and do not suppose a major risk for other health conditions. See line 281-285.